# The effects of pulmonary rehabilitation on inflammatory biomarkers in patients with chronic obstructive pulmonary disease: Protocol for a systematic review and meta-analysis

**Anastasia N. L. Newman**[1]*, **Ana Oliveira**[1,2,3]☯, **Roger Goldstein**[2,4,5,6]☯, **Christopher Farley**[1]☯, **Parameswaran Nair**[7]☯, **Dina Brooks**[1,2,4,5,6]☯

**1** Faculty of Health Science, School of Rehabilitation Science, McMaster University, Hamilton, ON, Canada, **2** Department of Respiratory Medicine, West Park Healthcare Centre, Toronto, ON, Canada, **3** Lab 3R – Respiratory Research and Rehabilitation Laboratory, School of Health Sciences (ESSUA) and Institute of Biomedicine, Aveiro, Portugal, **4** Faculty of Medicine, Department of Physical Therapy, University of Toronto, Toronto, ON, Canada, **5** Rehabilitation Sciences Institute, School of Graduate Studies, University of Toronto, Toronto, ON, Canada, **6** Faculty of Medicine, Department of Medicine, University of Toronto, Toronto, ON, Canada, **7** Department of Medicine, McMaster University, Hamilton, ON, Canada

☯ These authors contributed equally to this work.
* newmanan@mcmaster.ca

**Data Availability Statement:** No datasets were generated or analysed during the current study. All

## Abstract

### Introduction

Chronic obstructive pulmonary disease (COPD) is a common, preventable lung disease which affects more than 300 million people worldwide. People with COPD have elevated levels of inflammatory biomarkers, which are linked to physiological alterations in the respiratory system and extrapulmonary manifestations. Pulmonary rehabilitation (PR) is one of the strategies used in the management of individuals with COPD irrespective of severity, however its effect on systemic inflammation is poorly understood. We report the protocol of a systematic review on the effects of PR on systemic inflammation in patients with COPD.

### Materials and methods

Using the search terms "chronic obstructive pulmonary disease", "pulmonary rehabilitation", and "inflammatory biomarkers" and their synonyms, five databases (AMED, CINAHL, Ovid MEDLINE, MEDLINE (Pubmed), EMBASE) will be searched from their inception to identify primary literature evaluating the effects of PR on systemic inflammation. Two reviewers will independently screen titles, abstracts, and full texts for eligibility using the Covidence web-based software. Eligible studies must be published in a peer-reviewed journal and include: (1) participants with COPD undergoing PR with an exercise component of at least 4 weeks in length and (2) a measure of systemic inflammation (e.g., bloodwork or sputum sample) as an outcome of interest. We will use the Cochrane Risk of Bias Tools (ROB2 and ROBINS-I) and will rate the quality of the evidence using the Grading of Recommendations,

relevant data from this study will be made available upon study completion.

**Funding:** The authors received no specific funding for this work.

**Competing interests:** The authors have declared that no competing interests exist.

Assessment, Development and Evaluations (GRADE) tool. This protocol has followed the Preferred Reporting Items for Systematic Reviews and Meta-Analyses Protocols (PRISMA-P) guidelines and is registered with the International Prospective Register of Systematic Reviews (PROSPERO).

## Conclusion

The results of this systematic review will summarize the status of the evidence highlighting the effect of PR on systemic inflammation. A manuscript will be drafted and submitted to a peer-reviewed journal and shared at conferences.

## Introduction

Chronic obstructive pulmonary disease (COPD) affects more than 300 million people worldwide and is a leading cause of morbidity and mortality [1,2]. It is associated with primary respiratory impairments including dyspnea, coughing, sputum production, wheezing, and extrapulmonary systemic manifestations, such as muscle loss and cachexia, cardiovascular disease, osteoporosis, and metabolic syndromes (i.e., diabetes) [2,3]. Acute exacerbations of COPD (AECOPD) are common and include increased airway inflammation and mucous production, increased rates of hospitalization, and disease progression [2]. The World Health Organization (WHO) estimates that COPD accounts for six percent of all deaths (approximately three million people each year), and is one of the top three causes of mortality worldwide [2,4]. By 2060, it is expected that over 5.4 million annual deaths will be attributable to COPD due to a combination of an aging worldwide population and continued exposure to environmental irritants [2]. A diagnosis of COPD is also associated with a higher Disability-Adjusted Life Year (DALY) score, which calculates the sum of years lost due to premature mortality and years lived with disability [2,5]. Safiri et al noted that COPD accounted for approximately 74.4 million DALYs worldwide in 2019 alone [5]. In the European Union, the total direct costs of respiratory diseases are estimated to be six percent of the annual healthcare budget, with COPD accounting for 56% of all respiratory diseases (approximately 39 billion Euros per year) [2]. These costs are expected to increase worldwide over the next several decades, with the United States anticipating over 40 billion dollars per year to be spent on COPD management and hospital admissions [2].

Chronic inflammation is a key mediator in the pathophysiological changes associated with COPD, such as the remodeling and narrowing of the small airways and the alterations in both the lung parenchyma and the pulmonary vasculature [2,3,6]. This pathophysiology is also attributed to elevated levels of pro-inflammatory biomarkers, including macrophages, neutrophils, eosinophils, and lymphocytes and oxidative stress [2,3]. Oxidative biomarkers, including hydrogen peroxide, have been found in elevated concentrations in patients with COPD [2,3,6]. Levels of pro-inflammatory mediators, including cytokines, proteases, and chemokines, are also increased and attract inflammatory cells from the circulation, further modulating the inflammatory response [2,3]. Patients with persistent systemic inflammation have been found to have increased likelihood of developing muscle loss and cachexia, cardiovascular disease, osteoporosis, metabolic syndromes (i.e., diabetes), exacerbations, and have an increased risk of mortality [2,3]. The relationship between localized lung inflammation and systemic inflammation is poorly understood. It has been hypothesized that inflammation in the lung "overspills" into the circulation, leading to systemic inflammation [7]. This is supported by studies that

demonstrate the diffusion of inflammatory proteins from the lungs into peripheral circulation [7]. In a 2010 observational study, the authors noted that levels of inflammatory biomarkers peaked earlier in sputum versus blood samples [8]. However, these results are not consistent across the literature and further research is necessary to understand the link between lung and systemic inflammation.

Pulmonary rehabilitation (PR) is a cornerstone intervention for people with stable COPD and AECOPD [9–11]. It is defined as a comprehensive intervention based on thorough patient assessment followed by patient-tailored therapies that include, but are not limited to, exercise training, education, and self-management interventions aiming at behaviour change [2]. Pulmonary rehabilitation is designed to improve the physical and psychological condition of people with chronic respiratory diseases and to promote the long-term adherence to health-enhancing behaviours [2]. Evidence supports its efficacy for improving symptoms, exercise tolerance, and health-related quality of life in patients with both stable and exacerbated chronic respiratory diseases [9–14]. Previous literature has also suggested that acute exercise may be

**Table 1. PICO statement.**

| | |
|---|---|
| **Population** | Adults diagnosed with COPD using GOLD 2023 criteria:<br>• $FEV_1/FVC < 0.70$ via spirometry along with other signs:<br>• History of persistent, progressive dyspnea, often worse with exercise<br>Adults diagnosed with an AECOPD using GOLD 2023 definition:<br>• Acute worsening of respiratory symptoms that results in additional therapy (i.e., increased dyspnea, mucous production, coughing, wheezing)<br>• Other differential diagnoses ruled out (i.e., pulmonary embolus, pulmonary edema, pneumonia, pleural effusion) |
| **Intervention** | • Pulmonary rehabilitation (PR) using GOLD 2023 criteria:<br>• Comprehensive intervention based on thorough patient assessment • Includes exercise training, education, and self-management strategies<br>• For the purpose of this review, the included studies will require an exercise component for at least 4 weeks in length |
| **Comparator (RCTs)** | Non-PR control group or other interventions (i.e., education only) |
| **Outcomes of Interest** | Primary outcomes:<br>• IL-6<br>• Fibrinogen concentration (sodium citrate)<br>• C-reactive protein concentration<br>Secondary outcomes:<br>• Other cytokines (IL-8, IL-1, GM-CSF, IL-33, TNF-alpha)<br>• Enzyme-linked immunosorbent assay<br>• Proteins (albumin, ferritin, alpha-1 antitrypsin, club cell protein 16 (CCL16), surfactant protein-D (SP-D))<br>• Leukocyte count (neutrophil, eosinophil, monocytes, lymphocytes)<br>• Erythrocyte sedimentation rate (ESR)<br>• Other inflammatory markers (platelets, chemokines (PARC))<br>• Iron metabolism biomarkers<br>• Soluble receptor for advanced glycation products (sRAGE)<br>• Neutrophil elastase |
| **Setting** | • Outpatient rehabilitation setting (in-person)<br>• In-patient rehabilitation setting<br>• Home-based PR<br>• Tele-rehabilitation |
| **Study Designs** | • Prospective cohort studies (pre-post PR)<br>• Longitudinal observational studies (pre-post PR)<br>• Randomized controlled trials (non-PR control) |

GM-CSF = granulocyte-macrophage colony-stimulating factor; IL = interleukin; PARC = pulmonary and activation-regulated chemokine; TNF = tumor necrosis factor.

pro-inflammatory while regular, consistent exercise may promote an anti-inflammatory response [15]. However, the effects of PR on systemic inflammation in patients with COPD and AECOPD are not fully understood. The following is a protocol for a systematic review whose purpose is to summarize the poorly understood effects of PR on the systemic inflammatory response in patients with COPD.

### Research question

The primary research question for this systematic review is: What is the effect of pulmonary rehabilitation on systemic inflammation/systemic inflammatory biomarkers in patients with COPD or AECOPD? Further details about our PICO statement are provided in Table 1.

## Materials and methods

This review will follow the Cochrane methodology and the Preferred Reporting Items for Systematic Reviews and Meta-Analysis Protocols (PRISMA-P) reporting guidelines (S1 File) [16,17]. This review is registered with the International Prospective Register of Systematic Reviews (PROSPERO—CRD42023390089). Any changes to the protocol will be updated on PROSPERO.

### Eligibility criteria

Studies will be eligible for inclusion in this systematic review if they meet the following criteria: (1) published in a peer-reviewed journal, (2) original primary research (prospective cohort studies, longitudinal observational studies, and randomized controlled trials), (3) sample includes adults diagnosed with COPD or AECOPD enrolled in a PR program, (4) PR program includes an exercise component for at least four weeks, and (5) evaluates the effects of PR on the following inflammatory biomarkers: cytokines (e.g., IL-6, IL-8, TNF-alpha), fibrinogen concentration (sodium citrate), c-reactive protein concentration (CRP), leukocyte counts (e.g., neutrophil, eosinophil, monocytes, lymphocytes), erythrocyte sedimentation rate (ESR), proteins (e.g., albumin, ferritin, alpha-1 antitrypsin, club cell protein 16 (CCL16), surfactant protein-D (SP-D), and other inflammatory markers (e.g., platelets, chemokines, neutrophil elastase, soluble receptor for advanced glycation products (sRAGE)) and tests of inflammation (e.g., enzyme-linked immunosorbent assay).

### Search strategy

A detailed search strategy has been developed by the primary author (A.N.) and assisted by a health sciences librarian using key terminology for the population, intervention, and outcomes of interest. The search terms "chronic obstructive pulmonary disease", "pulmonary rehabilitation", and "inflammatory biomarkers" and their respective synonyms were used to search each database using Boolean operators. The search strategy for each database was validated by testing whether it could identify three relevant articles meeting our inclusion criteria previously identified through Google scholar. The search strategy is available on PROSPERO and in S2 File.

### Information sources

A comprehensive search of the following databases will be performed from their inception in Ovid MEDLINE, MEDLINE (Pubmed), EMBASE, CINAHL, and AMED to identify published peer-reviewed literature. References of eligible studies will be searched for any potentially relevant studies.

## Outcomes of interest

Our primary outcome of interest is the measurement of systemic inflammation through bloodwork or sputum samples, including IL-6, CRP, and fibrinogen concentrations. Secondary outcomes of interest include other inflammatory biomarkers such as ESR, protein concentrations, and leukocyte counts. A non-exhaustive list of possible inflammatory biomarkers is provided in Table 1.

## Data screening and extraction

Two reviewers working independently will screen titles, abstracts, and full texts of potential studies with discrepancies resolved by discussion. A third reviewer will be consulted if consensus cannot be reached. The same two reviewers will also perform data extraction independently and will review results to obtain consensus. Calibration exercises will be performed prior to initiating each stage of the review with six to eight papers. Screening will be performed using the Covidence online software and extraction will be performed using pre-defined tables in the Microsoft Excel Spreadsheet Software. The following information will be extracted from qualifying full texts:

a. Study identification (i.e., title, first author, year of publication, journal title, country of origin, trial registration number, study design, purpose/objectives, and funding sources)

b. Sample characteristics (i.e., size, age, sex, lung function, severity of the disease and severity of the airway obstruction, smoking history, co-morbid conditions (i.e., diabetes mellitus, cardiovascular disease, rheumatoid conditions, etc.))

c. Participants use of corticosteroids in relation to timing of bloodwork (i.e., inhaled oral, or burst deliveries)

d. Concomitant medications used by participants (i.e., lipid, cholesterol, and glucose modifying drugs)

e. Information on participants dietary habits or caloric intake (as part of PR or otherwise)

f. PR program characteristics (i.e., components, sessions duration and frequency, setting and length)

g. Outcome(s) assessed (i.e., specific inflammatory biomarker assessed)

h. Timing of the outcome(s) of interest assessments

i. Adherence to PR program

j. Information about control group (if applicable) (i.e., sample size, age, sex)

k. Mean difference (standard deviation) or median (interquartile ranges) of change in bloodwork/sputum samples

l. Summary of results of each included study

## Risk of bias assessment

Two reviewers will independently assess the risk of bias of all included studies. The Cochrane Risk of Bias Tool for Randomized Trials (ROB2) will be used for all included randomized controlled trials [18]. This tool assesses the risk of bias arising from the randomization process, deviations from the intended intervention, missing outcomes, measurement of the outcome,

and selection of the reported results. For non-randomized trials, the Cochrane Risk of Bias In Non-Randomized Studies–of Interventions (ROBINS-I) tool will be employed [19]. This tool also evaluates the inherent risk of bias using specific criteria (i.e., bias due to: confounding, selection of participants, classification of interventions, deviations from intended interventions, missing data, measurement of outcomes, and selection of the reported results). A summary of both the ROB2 and the ROBINS-I results and their impact on the interpretation of the results will be provided.

### Quality of evidence

The Grading of Recommendations, Assessment, Development and Evaluations (GRADE) will be utilized to grade the certainty or quality of the evidence and strength of the recommendation [20]. Two investigators will score all included studies using the GRADE criteria. Disagreements will be resolved via discussion with a third reviewer available if consensus cannot be achieved. This will be performed for the primary outcomes of interest (listed in Table 1).

### Data analysis and synthesis

A descriptive summary of the effects of PR and systemic inflammation in people with COPD will be performed. This information will be presented in the body of the manuscript and in tables throughout the review. Statistical results, including mean differences (standard deviation) for parametric data and medians (interquartile range) for nonparametric data, will be collected. Excel will be used for data analysis. Heterogeneity of the data will be assessed using the $I^2$ statistic and values above 50% will be identified as having substantial heterogeneity. Statistical significance will be set at $p < 0.05$. For similar outcomes of interest, meta-analyses will be conducted using the ReviewManager (RevMan, version 5.4.1). All study related data will be made freely available on Open Science Framework at the time of publication.

### Dissemination of results

The results of this systematic review will be submitted for presentation at a respiratory-relevant conference. A manuscript will be written and submitted for consideration of publication in a peer-reviewed journal.

### Significance

Both the pulmonary and non-pulmonary manifestations of COPD have been associated with elevated levels of systemic inflammation [2,9–11]. PR is a cornerstone treatment for individuals diagnosed with COPD, however its impact on systemic inflammation is poorly understood. This review will summarize the available primary evidence investigating the effects of PR on inflammatory biomarkers, to help clarify the direction of these effects, and to summarize the strength and quality of the evidence available. The results could be used to determine what modifications, if any, may be considered for PR programming based on the systemic inflammatory response post-PR.

### Supporting information

**S1 File. PRISMA-P checklist.**
(DOC)

**S2 File. Search strategy.**
(DOCX)

## Acknowledgments

The authors would like to thank Mr. Jack Young, Health Sciences Librarian at McMaster University, for his assistance with designing the search strategies.

## Author Contributions

**Conceptualization:** Anastasia N. L. Newman, Ana Oliveira, Roger Goldstein, Parameswaran Nair, Dina Brooks.

**Formal analysis:** Anastasia N. L. Newman, Roger Goldstein.

**Investigation:** Parameswaran Nair.

**Methodology:** Anastasia N. L. Newman, Ana Oliveira, Roger Goldstein, Christopher Farley, Parameswaran Nair, Dina Brooks.

**Project administration:** Anastasia N. L. Newman, Dina Brooks.

**Resources:** Christopher Farley.

**Software:** Christopher Farley.

**Supervision:** Dina Brooks.

**Validation:** Anastasia N. L. Newman, Ana Oliveira, Christopher Farley.

**Writing – original draft:** Anastasia N. L. Newman, Dina Brooks.

**Writing – review & editing:** Anastasia N. L. Newman, Ana Oliveira, Roger Goldstein, Christopher Farley, Parameswaran Nair, Dina Brooks.

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
