## [Decision Letter · Decision Letter 0]

24 Apr 2023

PONE-D-23-06071The effects of pulmonary rehabilitation on inflammatory biomarkers in patients with chronic obstructive pulmonary disease: Protocol for a systematic reviewPLOS ONE

Dear Dr. Newman,

Thank you for submitting your manuscript to PLOS ONE. After careful consideration, we feel that it has merit but does not fully meet PLOS ONE’s publication criteria as it currently stands. Therefore, we invite you to submit a revised version of the manuscript that addresses the points raised during the review process.

We look forward to receiving your revised manuscript.

Kind regards,

Felix Bongomin, MB ChB, MSc, MMed, FECMM

Academic Editor

PLOS ONE

Journal Requirements:

Additional Editor Comments:

Please revise the manuscript as suggested by the reviewer and adhere to all of the items in the MOOSE/PRISMA-P guidelines

Reviewers' comments:

Reviewer's Responses to Questions

**Comments to the Author**

1. Does the manuscript provide a valid rationale for the proposed study, with clearly identified and justified research questions?

Reviewer #1: Partly

2. Is the protocol technically sound and planned in a manner that will lead to a meaningful outcome and allow testing the stated hypotheses?

Reviewer #1: Yes

3. Is the methodology feasible and described in sufficient detail to allow the work to be replicable?

Reviewer #1: Yes

4. Have the authors described where all data underlying the findings will be made available when the study is complete?

Reviewer #1: No

5. Is the manuscript presented in an intelligible fashion and written in standard English?

Reviewer #1: Yes

6. Review Comments to the Author

You may also provide optional suggestions and comments to authors that they might find helpful in planning their study.

Reviewer #1: Thank you for the paper. It is very enlightening, but I have a few comments for the authors that they could consider.

Data availability statement:

Since this is a protocol, this statement “No datasets were generated or analysed during the current study” is not applicable to the study design. The authors should specifically highlight which platform will be used to share the data used in this study.

Abstract

Introduction

Line 38- 39: The two sentences can be joined to form one. I suggest that you replace cornerstone treatment with “one of the strategies used in the management of individuals with COPD irrespective of severity but its effect on systemic inflammation is poorly understood.”

Still in the abstract: the rationale for conducting the systematic review is not clear. Is it because the effect is poorly understood or because there is no previous systematic review highlighting the effect?

Materials and methods:

Line 45: What do you mean from inception?

Line 49: The inclusion criteria 1 is not clear. “PR with an exercise component of at least 4 weeks “or I would rather suggest that the authors edit it to make it clearer.

Line 56: Conclusion not conclusions

Line 57- 59: I suggest the authors considered rewording the sentence to “The results from this systematic review will summarize the status on evidence highlighting/ showing/ illustrating the effect of PR on systemic inflammation and a manuscript will be drafted and submitted to a peer-reviewed journal or shared at conferences.”

Main text

Introduction

Line 65 to 73: There is no need to define what COPD or Acute Exacerbation of COPD is, it is common knowledge. The authors should highlight the burden then move on to how systemic inflammation comes into the picture.

Line 74 to 85: COPD is a product of localized inflammation in lungs which is supported by the pathological changes highlighted in line 75 and 76. I suggest that the authors highlight how this localized inflammation becomes systemic inflammation. Authors should also know that biomarkers are proxies of the pathological state of the organs case in point the lungs. The authors should consider highlighting biomarkers that are highly specific to COPD or AECOPD and not all inflammatory biomarkers as these might be associated with other diseases.

Line 86- 95: The authors highlighted what PR is. The rationale for conducting the systematic review is not clear.

Methods

Under intervention:

“the included studies will have an exercise component for at least 4 weeks in length.”

Outcome of interest:

“Enzyme-linked immunosorbent assay” this is not a marker rather a test.

Could the authors focus on COPD specific biomarkers? Or provide a justification as to why they are considering all these biomarkers in the introduction that appear nonspecific?

Line 110- 111: Please include a reference for the reporting guidelines.

Line 120: Instead of “of” consider “for.”

Line 157: In the data extraction: The authors should include existing co-morbidities focusing on diseases that induce chronic inflammation or that contribute to high concentrations of inflammatory biomarkers.

Line 206: The authors report that meta-analyses will be conducted where possible. Why bring this up now? Why aren’t they doing a meta-analysis in the first place?

Line 213: The authors could change conclusion to “Significance.”

7. PLOS authors have the option to publish the peer review history of their article (what does this mean?). If published, this will include your full peer review and any attached files.

Reviewer #1: No

---

## [Author Response · Author response to Decision Letter 0]

21 May 2023

May 19, 2023

Dr. Felix Bongomin, MB ChB

Academic Editor, PLOS One

Dear Dr. Bongomin,

RE: The effects of pulmonary rehabilitation on inflammatory biomarkers in patients with chronic obstructive pulmonary disease: Protocol for a systematic review and meta-analysis

Reference Number: PONE-D-23-06071

Principal Author: Dr. Anastasia Newman, MSc(PT), MSc(RS), PhD

Thank you for reviewing our original research article, entitled “The effects of pulmonary rehabilitation on inflammatory biomarkers in patients with chronic obstructive pulmonary disease: Protocol for a systematic review”, and providing comments and feedback. We are grateful for the opportunity to clarify our research and address the reviewer’s questions. We address each of the reviewer’s comments in the response below.

Journal Requirements:

J1C1: Please ensure that your manuscript meets PLOS ONE’s style requirements, including those for file naming. 

J1C1 Response: Thank you for highlighting the style discrepancies. We have updated the manuscript to align with PLOS ONE’s style instructions.

Reviewer 1:

R1C1: Since this is a protocol, this statement “No datasets were generated or analysed during the current study” is not applicable to the study design. The authors should specifically highlight which platform will be used to share the data used in this study. 

R1C1 Response: Thank you for this comment. We have modified the manuscript as follows:

“All study related data will be made freely available on Open Science Framework at the time of publication.” (Page 13)

R1C2: Line 38-39: The two sentences can be joined to form one. I suggest that you replace cornerstone treatment with “one of the strategies used in the management of individuals with COPD irrespective of severity but its effect on systemic inflammation is poorly understood”

R1C2 Response: Thank you for this comment. The text has been modified to reflect the suggested changes:

“Pulmonary rehabilitation (PR) is one of the strategies used in the management of individuals with COPD irrespective of severity, however its effect on systemic inflammation is poorly understood.” (Page 2)

R1C3: The rationale for conducting the systematic review is not clear. Is it because the effect is poorly understood or because there is no previous systematic review highlighting the effect?

R1C3 Response: Thank you for this feedback. We have removed the sentence “To our knowledge, there has been no previous systematic review on this topic” in abstract. The rationale for this systematic review is to summarize the effects of pulmonary rehabilitation on systemic inflammation in people with COPD as these effects are not fully understood.

R1C4: Line 45: What do you mean from inception?

R1C4 Response: We have added the word “their” to help clarify the statement that each database will be searched from their initial creation until the date the search was conducted.

“Using the search terms “chronic obstructive pulmonary disease”, “pulmonary rehabilitation”, and “inflammatory biomarkers” and their synonyms, five databases (AMED, CINAHL, Ovid MEDLINE, MEDLINE (Pubmed), EMBASE) will be searched from their inception to identify primary literature evaluating the effects of PR on systemic inflammation.” (Page 2)

R1C5: Line 49: The inclusion criteria 1 is not clear. “PR with an exercise component of at least 4 weeks” or I would rather suggest that the authors edit it to make it clearer.

R1C5 Response: We have added the following to clarify the exercise component:

“(1) participants with COPD undergoing PR with an exercise component of at least 4 weeks in length…” (Page 2)

R1C6: Line 56: Conclusion not conclusions.

R1C6 Response: We have made the change as suggested. Thank you. 

R1C7: Line 57-59: I suggest the authors considered rewording the sentence to “The results from this systematic review will summarize the status on evidence highlighting/showing/illustrating the effect of PR on systemic inflammation and a manuscript will be drafted and submitted to a peer-reviewed journal or shared at conferences.”

R1C7 Response: Thank you for this feedback. We have made the suggested changes:

“The results of this systematic review will summarize the status of the evidence highlighting the effect of PR on systemic inflammation. A manuscript will be drafted and submitted to a peer-reviewed journal and shared at conferences.” (Page 3)

R1C8: Line 65-73: There is no need to define what COPD or Acute Exacerbation of COPD is, it is common knowledge. The authors should highlight the burden then move on to how systemic inflammation comes into the picture.

R1C8 Response: We have removed some of the text as requested and have elaborated on the worldwide burden of COPD from a socioeconomic, mortality, and morbidity perspectives.

R1C9: Line 74-85: COPD is a product of localized inflammation in lungs which is supported by the pathological changes highlighted in line 75 and 76. I suggest that authors highlight how this localized inflammation becomes systemic inflammation. Authors should also know what biomarkers are proxies of the pathological state of the organ case in point the lungs. The authors should consider highlighting biomarkers that are highly specific to COPD and AECOPD and not all inflammatory biomarkers as these might be associated with other diseases.

R1C9 Response: Thank you for this feedback. The evidence to support the transition from local to systemic inflammation remains preliminary. We have added a brief overview of the suspected pathophysiological mechanism linking lung inflammation and systemic inflammation. 

Unfortunately, the evidence isn’t clear as to which inflammatory biomarkers are specific proxies for lung inflammation. As such, we are not comfortable limiting our search at this time. Please see our response with respect to our selection of biomarkers and rationale for including them below (R1C13).

R1C10: Line 86-95: The authors highlighted what PR is. The rationale for conducting the systematic review is not clear.

R1C10 Response: We hope the following changes clarify the purpose of our systematic review:

“However, the effects of PR on systemic inflammation in patients with COPD and AECOPD are not fully understood. The following is a protocol for a systematic review whose purpose is to summarize the poorly understood effects of PR on the systemic inflammatory response in patients with COPD.” (Page 6)

R1C11: Under intervention: “the included studies will have an exercise component for at least 4 weeks in length.”

R1C11 Response: We have altered the statement as suggested. Thank you.

“For the purpose of this review, the included studies will require an exercise component for at least 4 weeks in length.” (Page 7)

R1C12: Outcome of interest: “Enzyme-linked immunosorbent assay” this is not a marker rather a test.

R1C12 Response: Thank you identifying this inaccuracy. We have removed the statement “Measures of systemic inflammation (blood tests)” and simply listed all the inflammatory markers/tests of interest. (Page 7)

R1C13: Could the authors focus on COPD specific biomarkers? Or provide a justification as to why they are considering all these biomarkers in the introduction that appear non-specific?

R1C13 Response: The reviewer has raised a pertinent concern that is relevant to the selection of biomarkers associated with COPD. Unfortunately, no biomarker has been evaluated to have perfect diagnostic, theragnostic, or predictive properties for COPD exacerbations, progression, or response to anti-inflammatory treatment, perhaps with the exception of alpha-1 anti-trypsin deficiency, which is not expected to improve with pulmonary rehabilitation. This concern has been reviewed recently by Stockley, Halpin, Celli and Singh (Am J Respir Crit Care Med 2019; 199: 1195-1204). Therefore, we are not confident that we could limit our analysis to specific markers in the absence of their association with COPD being more reliably established.

The four most consistently observed in association with COPD exacerbations in blood are: IL-6 (non-specific), CRP (non-specific), fibrinogen (non-specific) and eosinophilia (non-specific). Additional biomarkers that are more specific for the lung include: CCL16 (Club Cell Protein), SP-D (Surfactant Protein-D), sRAGE (soluble Receptor for Advanced Glycation Products), and Neutrophil elastase.

While CCL16, SP-D and sRAGE might have anti-inflammatory properties (or act as an inflammation decoy in the case of sRAGE), the others could be non-specific acute phase reactants. The role of blood eosinophils in COPD is not well established. While it is associated with response to treatment with inhaled or oral corticosteroids and with exacerbations, for most patients it may not be directly involved in the pathobiology of disease.

R1C14: Line 110-111: Please include a reference for the reporting guidelines.

R1C14 Response: Thank you for identifying this omission. We have added the references for the PRISMA-P reporting guidelines. 

R1C15: Line 120: Instead of “of” consider “for”.

R1C15 Response: We have made the edit as requested.

R1C16: Line 157: In the data extraction: The authors should include existing co-morbidities focusing on diseases that induce chronic inflammation or that contribute to high concentrations of inflammatory biomarkers.

R1C16 Response: Thank you for this feedback. We have added the following statement to the list of data to be extracted:

“Sample characteristics (i.e., size, age, sex, lung function, severity of the disease and severity of the airway obstruction, smoking history, co-morbid conditions (e.g., diabetes mellitus, cardiovascular disease, rheumatoid conditions, etc…))” (Page 11)

R1C17: Line 206: The authors report that meta-analyses will be conducted where possible. Why bring this up now? Why aren’t they doing a meta-analysis in the first place?

R1C17 Response: Thank you for this comment. It was always our intention to complete a meta-analysis when homogeneity of outcomes permitted. We realize this may not have been clearly stated in the manuscript and have made changes based on the feedback. We have removed “Whenever possible” from the sentence to strengthen our statement that meta-analyses will be conducted:

“For similar outcomes of interest, meta-analyses will be conducted using the ReviewManager (RevMan version 5.4.1).” (Page 13)

R1C18: Line 213: The authors could change conclusion to “Significance.”

R1C18 Response: We have made the change from CONCLUSION to SIGNIFICANCE as requested.

Thank you to the reviewer and the Editor for their feedback. We look forward to hearing from PLOS One and hope these revisions now provide a manuscript acceptable for publication. 

Regards,

Anastasia Newman

MSc(PT), MSc(RS), PhD Candidate

---

## [Decision Letter · Decision Letter 1]

7 Jun 2023

The effects of pulmonary rehabilitation on inflammatory biomarkers in patients with chronic obstructive pulmonary disease: Protocol for a systematic review and meta-analysis

PONE-D-23-06071R1

Dear Dr. Newman,

We’re pleased to inform you that your manuscript has been judged scientifically suitable for publication and will be formally accepted for publication once it meets all outstanding technical requirements.

Kind regards,

Felix Bongomin, MB ChB, MSc, MMed, FECMM

Academic Editor

PLOS ONE

Additional Editor Comments (optional):

Reviewers' comments:

Reviewer's Responses to Questions

**Comments to the Author**

1. Does the manuscript provide a valid rationale for the proposed study, with clearly identified and justified research questions?

Reviewer #1: Yes

2. Is the protocol technically sound and planned in a manner that will lead to a meaningful outcome and allow testing the stated hypotheses?

Reviewer #1: Yes

3. Is the methodology feasible and described in sufficient detail to allow the work to be replicable?

Reviewer #1: Yes

4. Have the authors described where all data underlying the findings will be made available when the study is complete?

Reviewer #1: Yes

5. Is the manuscript presented in an intelligible fashion and written in standard English?

Reviewer #1: Yes

6. Review Comments to the Author

You may also provide optional suggestions and comments to authors that they might find helpful in planning their study.

Reviewer #1: Abstract

Line 24: I suggest the authors remove “common.”

Line 27: I think this sentence is incomplete “and extrapulmonary manifestations of…….” What is the author trying to say/ write?

Introduction:

Line 57- 59: Is there a difference between that sentence and the sentence in line 51- 52? I think they all mean the same. I suggest that the authors delete one.

I further suggest that the authors re-align the first paragraph. Here is my recommendation:

Line 51 to 52 open then paragraph. This could be followed by sentences from line 60 to 69.

The sentences which start from line 52 “It is associated with primary respiratory impairments including dyspnea, coughing, sputum production, wheezing, and extrapulmonary systemic manifestations, such as muscle loss and cachexia, cardiovascular disease, osteoporosis, and metabolic syndromes (i.e., diabetes) to line 55 can be rephrase and start second the paragraph. This could make it easy to follow and so that we do not have a mix of symptoms and statistics.

Another suggestion to the authors for their consideration. Since there is no specific biomarker to show the effect of PR on systemic inflammation, you could mention that the meta-analysis could provide evidence to build on this assumption.

7. PLOS authors have the option to publish the peer review history of their article (what does this mean?). If published, this will include your full peer review and any attached files.

Reviewer #1: **Yes: **Andrew Weil Semulimi

---

## [Editor Report · Acceptance letter]

16 Jun 2023

PONE-D-23-06071R1 

The effects of pulmonary rehabilitation on inflammatory biomarkers in patients with chronic obstructive pulmonary disease: Protocol for a systematic review and meta-analysis 

Dear Dr. Newman:

I'm pleased to inform you that your manuscript has been deemed suitable for publication in PLOS ONE. Congratulations! Your manuscript is now with our production department. 

Kind regards, 

on behalf of

Dr. Felix Bongomin 

Academic Editor

PLOS ONE